# Hydrophobic-hydrophilic crown-like structure enables aquatic insects to reside effectively beneath the water surface

Chiaki Suzuki[1,8], Yasuharu Takaku [1,2,8✉], Hiroshi Suzuki[3], Daisuke Ishii[4], Tateo Shimozawa[5], Shuhei Nomura[6], Masatsugu Shimomura[7] & Takahiko Hariyama [1,2✉]

Various insects utilise hydrophobic biological surfaces to live on the surface of water, while other organisms possess hydrophilic properties that enable them to live within a water column. Dixidae larvae reside, without being submerged, just below the water surface. However, little is known about how these larvae live in such an ecological niche. Herein, we use larvae of *Dixa longistyla* (Diptera: Dixidae) as experimental specimens and reveal their characteristics. A complex crown-like structure on the abdomen consists of hydrophobic and hydrophilic elements. The combination of these contrasting features enables the larvae to maintain their position as well as to move unidirectionally. Their hydrophobic region leverages water surface tension to function as an adhesive disc. By using the resistance of water, the hydrophilic region serves as a rudder during locomotion.

[1] Preeminent Medical Photonics Education & Research Center, Institute for NanoSuit Research, Hamamatsu University School of Medicine, Higashi-ku, Hamamatsu, Japan. [2] NanoSuit Inc., Higashi-ku, Hamamatsu, Japan. [3] Department of Chemistry, Hamamatsu University School of Medicine, Higashi-ku, Hamamatsu, Japan. [4] Life Science and Applied Chemistry, Graduate School of Engineering, Nagoya Institute of Technology, Gokiso-cho, Showa-ku, Nagoya, Japan. [5] Research Institute for Electronic Science, Hokkaido University, N21W10, Kita-ku, Sapporo, Japan. [6] National Museum of Nature and Science, Tsukuba, Japan. [7] Chitose Institute of Science and Technology, Departments of Bio- and Material Photonics, Chitose, Japan. [8] These authors contributed equally: Chiaki Suzuki, Yasuharu Takaku. ✉email: ytakaku@hama-med.ac.jp; hariyama@hama-med.ac.jp

An important environmental adaptation of aquatic organisms is the wettability of their body surface. This controls the niche they occupy and plays an important role in their survival strategy. Several studies have focused on how various walking insects and spiders reside on the surface of water[1,2]. Water striders have remarkable water-repellent legs that enable them to walk on the water surface using the surface tension effect. The striders move quickly on the water by a sculling motion, whereby momentum is transferred by U-shaped vortices dominated by the water surface tension[3]. These organisms achieve hydrophobicity by chemical modifications with wax or oil secreted on their outer surface and/or fine hairy structures[4].

By contrast, other aquatic organisms possess hydrophilic properties that enable them to live within a water column. For example, the feeding habits of various diving beetles in water have been investigated[5,6]. Furthermore, the floating for respiration has been examined in *Aedes* mosquito larvae in water[7]. Anopheline larvae usually dive under water and remain at the bottom for some time. This pattern of behaviour suggests that their diving ability may be related to feeding and predator avoidance[8].

In contrast to these organisms, the larvae of Dixidae, a family of aquatic Nematocera flies (Diptera), exhibit a unique characteristic[9,10]. The larvae of Dixidae usually live along the margin of floating branches, stalks of waterweed, rocks in a pond, or parts of a stream where the current is very gentle. Like mirror images of water striders, the larvae of Dixidae lie with their ventral surface just under the water surface. They never dive into the water and are never completely submerged within the water column; therefore, they are known as meniscus midges. Little is known, however, about how the larvae are able to live beneath the water surface. In this study, we used larvae of *Dixa longistyla* (Diptera: Dixidae) as experimental specimens. We examined their ability to live just below the water surface, focusing on crown-like structures on the ventral side of the hind segments[10]. We provide evidence that a combination of hydrophobic and hydrophilic structures enables the larvae to reside in such an ecological niche.

## Results

**Confirmation of hydrophobic–hydrophilic properties.** In the larvae of Dixidae, the ventral side of the hind segments consists of five unique crown-like structures (referred to as crown hereafter) adhering to the water surface (Fig. 1a–c, arrowheads) and a terminal complex (Fig. 1a–c, arrow) (Supplementary Movie 1). High-magnification observations with light microscopy show hairy structures in the crowns. They further reveal that the tips of the hairs repel water and are connected to the air by point contacts (Figs. 1b, inset, 2a–c, e), suggesting that the tips of the hairs are hydrophobic[11,12]. In contrast, the other regions of the crown hairs are in direct contact with water and hence demonstrate hydrophilicity.

Observations with electron microscopy (Supplementary Movie 2)[13–16] show that the crowns consist of three different structures (Fig. 2g–l and Supplementary Fig. 1). In the centre (region 1), flat leaf-like structures ~3–5 μm in width are observed with central folds (Fig. 2i). Surrounding the central structures (region 2), dense hairs (microtrichia) <0.3–1 μm in diameter are detected (Fig. 2j). On the outer periphery (region 3), sparse hairs (protrusion) ~0.5–2 μm in diameter are observed (Fig. 2k). In regions 2 and 3, the tips of the hairs are further divided into branched structures and can be distinguished by their diameter of 0.1 μm (Fig. 2d, j) or 0.3 μm (Fig. 2f, k), respectively. These hairy structures are not hard spines; rather, they are observed as flexible in water and air (Fig. 3). All five crowns show the same configuration consisting of the three regions.

To quantify its surface affinity to water, we investigated the wettability of the crown using a microscopic contact angle metre

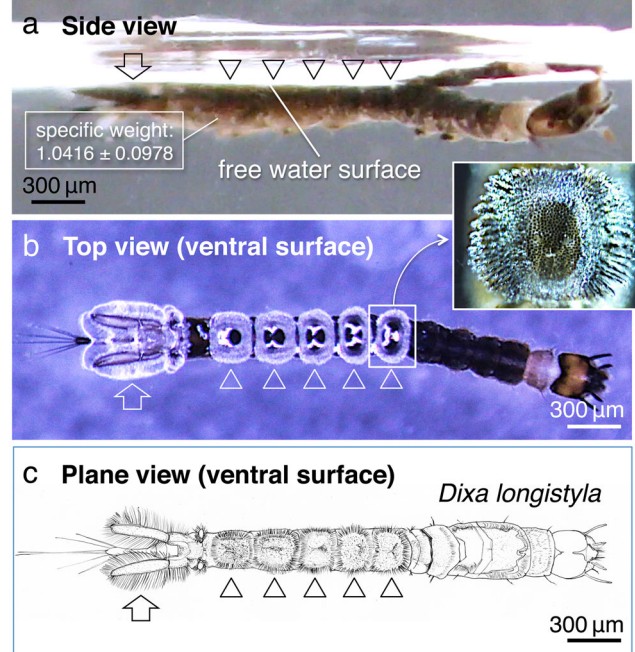

**Fig 1 Observations of the *Dixa longistyla* larvae. a** Side-view and **b** top-view stereo dissecting microscopy images. **c** Schematic of the larva. The arrowheads and arrows indicate the positions of the five crown-like structures and the terminal complex of the body segment, respectively.

(Fig. 2i–k, insets)[17]. In region 1, the water contact angle ($\theta$) is almost 0°, indicating super-hydrophilic properties, while the tips of microtrichia in region 2 are superhydrophobic ($\theta$ = ~150°). The angle $\theta$ = 110° was measured for the tips of protrusions in region 3, indicating hydrophobic properties, whereas the body of the animal adjacent to the crowns was found to be slightly hydrophilic with $\theta$ = 50° (region 4) (Fig. 2l, inset). These results indicate that the crown has a complex hydrophobic–hydrophilic structure (Fig. 2g, h).

**Role of the hydrophobic crown structure.** To analyse the mechanism by which *Dixa* are maintained under the surface, we attempted to submerge the larvae with a pair of forceps (Fig. 4a and Supplementary Movies 3 and 4). This was difficult because the larvae were tightly attached to the water surface; however, some specimens were eventually forcibly pulled under and sunk below the water surface. In these experiments, the terminal complex was always the first to re-attach to the water surface. However, this terminal complex re-attachment was insufficient to return the larva to its natural horizontal position parallel to the water surface (Fig. 4a '2.0 s'). In contrast, when the five crowns were re-attached to the water surface, the horizontal position was stably re-established (Fig. 4a '7.5 s'). These results led to the hypothesis that, although the terminal complex of the segments is a buoyant structure, the crown has a different function. By utilising the interfacial tension between the crowns and the air–water interface, the crowns fix the larva to its horizontal position below the surface.

To address this issue, we designed a simple experiment. When the liquid surface tension was reduced, the crown structures lost contact with the water surface, and the position of the larvae quickly changed from being parallel to being perpendicular to the surface (Fig. 4b). Typically, when these larvae swim, the body moves forward, parallel to the surface; however, in this case, the direction of motion was perpendicular and into the water column. We also found that, although the specific weight of the

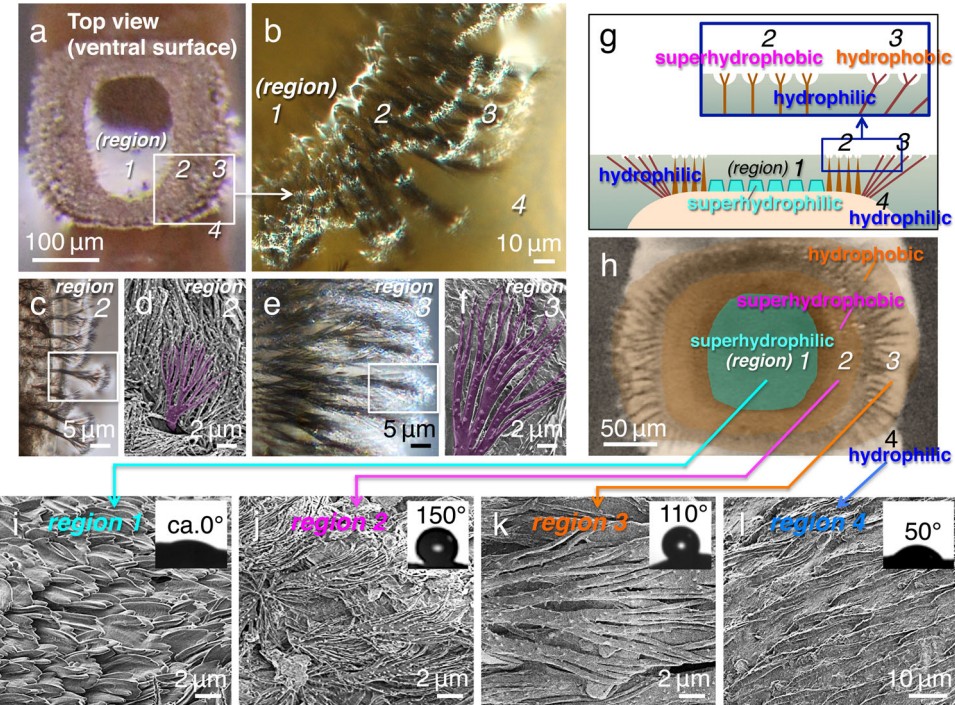

**Fig 2 Observations of the crown-like structures in the larvae. a** Stereo dissecting microscopy and **b** differential interference microscopy images of the crown. **c, e** High-magnification differential interference microscopy images of the tips of the hairy crown structures underneath the water surface in regions 2 in **c** and 3 in **e**. **d, f** SEM images of areas corresponding to the white rectangles in **c** and **e**, respectively, obtained from samples prepared via the NanoSuit method for living specimens. **g, h** Schematic images showing four regions with different wettabilities (1-4): cross-sectional view in **g** and top view in **h**. **i–l** SEM images of these regions in specimens prepared using the NanoSuit method. Water contact angle measurements are shown in the insets.

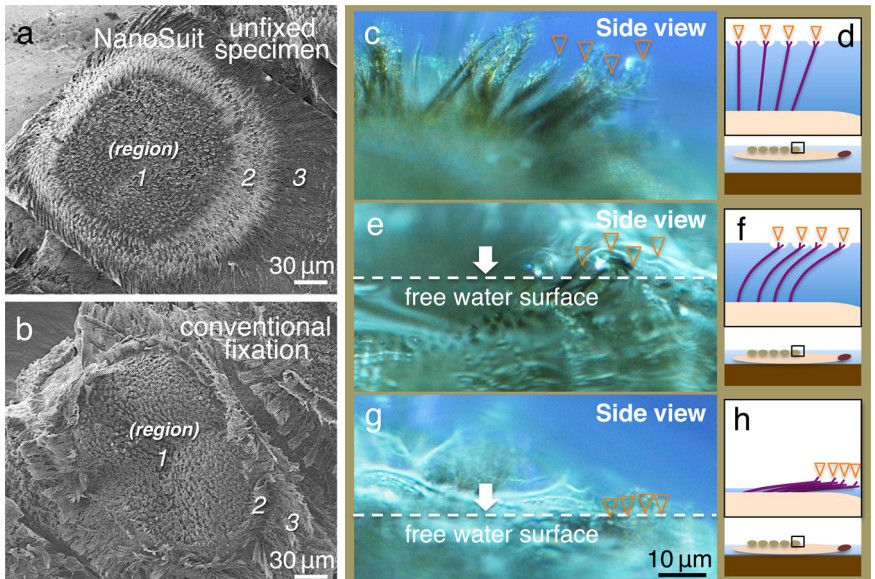

**Fig 3 Hairy structures in the crown show flexible property. a, b** SEM images of the crown structure treated either using the NanoSuit method in **a** or prepared by conventional fixation methods in **b**. Note that for the sample prepared using the NanoSuit living-specimen method, upon being pulled out of the water, hairy structures were observed to droop down, suggesting that they were flexible. In contrast, for the sample prepared using conventional fixation methods, the hairy structures appeared to have been dried out by the various pre-treatments. **c, e, g** Optical microscopy and **d, f, h** corresponding schematic images of the hairy structures in the crown responding to the water surface at different heights. The overlaid orange arrowheads indicate the positions of the tips of the hairy structures. The flexible structures bend when the surface of the water is below the tips, and the underside of the body is in contact with the bottom of the container.

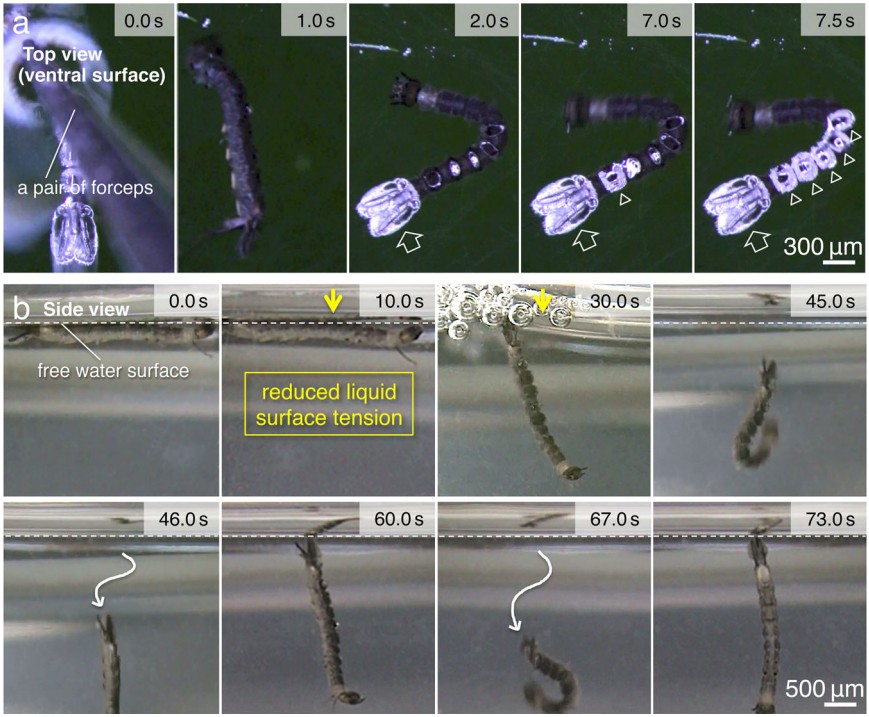

**Fig 4 Analysis of the interaction between the liquid surface tension and the crowns. a** Time-lapse top-view video image sequence of the larva captured during and after it was forcefully sunk into water with a pair of forceps. **b** Side-view images of the larva, before and after reducing the liquid surface tension, showing the postural change of the larva from being parallel (10.0 s) to perpendicular (30.0 s) with respect to the water surface.

animal was measured as ~1.04 (Fig. 1a), the specimens were always able to float again, perpendicular to the water surface, with the uppermost terminal complex of the segments. When the experimental larvae were transferred back into the water with the natural liquid surface tension, they reassumed their normal position, parallel to the surface (Fig. 1). The results suggest that the hairy tips of the hydrophobic crown utilise surface tension, making it strongly adherent when parallel to the water surface. Video images of spontaneous movements of the larva, captured from a side view, further support this interpretation of the experimental observations (Supplementary Movie 5).

**Pull-off experiments**. To better understand the relationship between the crowns and the liquid–air interfacial tension, we performed experiments in which the crowns were pulled away from the surface. When the crowns attached to the surface were pulled perpendicularly into the water, the water surface curved down with the crowns (Fig. 5a, b and Supplementary Movies 6 and 7). During these experiments, the tips of the hydrophobic hairy structures around the periphery of the crown (regions 2 and 3) displayed the strongest capacity to repel the water surface.

To confirm the role of the tension at the liquid interface, we constructed artificial substrates that mimicked the wettability of the crown. We conducted experiments that involved pulling the artificial ring substrates under water (Fig. 5c, d). We tested rings with both hydrophobic and hydrophilic surfaces[17]. The hydrophobic rings had a water contact angle ($\theta$) of 120°, whereas $\theta = 40°$ for the hydrophilic rings. When the rings were pulled into the water, the water surface initially curved down and then detached from the rings. The distance from the free water surface was determined at the moment of detachment. The maximum distances were significantly higher for the hydrophobic substrates ($3.4 \pm 0.1$ mm, mean ± S.D., $N = 10$) (Fig. 5c) than for the hydrophilic substrates ($2.8 \pm 0.1$ mm, mean ± S.D., $N = 10$) ($P < 0.01$) (Fig. 5d). Taken together, these results suggest that the

hydrophobic property of the peripheral region works for sticking the crowns to the water surface effectively.

**Role of the hydrophilic crown structure in swimming behaviour**. Finally, to determine the functions of their horizontal position below the water surface, we investigated the swimming movements of larvae (Figs. 6 and 7). In these experiments, we compared the larvae of *Dixa longistyla* that we had used as control samples (Fig. 6d, e) with larvae of *Dixella subobscura*, which belong to the same Dixidae family and do not have the crown structures on the ventral side of the hind segments (Fig. 6a, b). When these larvae were surprised or moved spontaneously, they promptly swam under the water surface. However, we found that, although the *Dixella* larvae showed random movement directions (Fig. 6c and Supplementary Movie 8), the *Dixa* larvae swam in an almost straight line (Fig. 6f and Supplementary Movie 9). During the movements, the *Dixella* larvae showed incomplete contact with the water surface (Fig. 6b), and occasionally, the bodies of the larvae were perpendicular to the surface (Supplementary Movie 8). This result suggests that the crown structures contribute to maintaining their horizontal position; therefore, they can regulate the unidirectional movement. Moreover, to propel themselves forward in the swimming movements, the animals showed horizontal left–right waving in their anterior body segments, while their posterior segments remained straight. This observation prompted our hypothesis that the unidirectional movement may be dependent on the hydrophilic property of the crown (Fig. 7a–c). Such hydrophilic property enhances the resistance of water and hence enables the posterior segments to be stable under the water surface.

To investigate whether the swimming movements were controlled by the wettability of the crowns, we treated the hydrophilic region with numerous hydrophobic oils. The best results were obtained with sunflower seed oil, which is a non-toxic and non-volatile organic substance commonly used in food or cosmetic

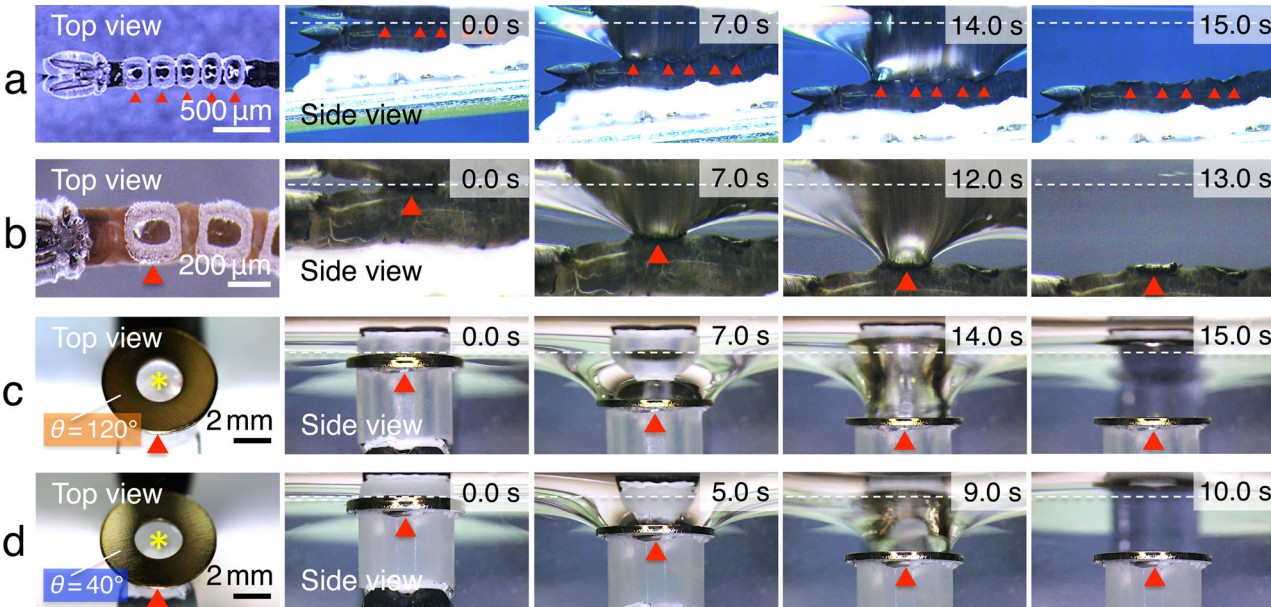

**Fig 5 Pull-off experiments from the water surface. a, b** Top- and side-view video images of the larvae acquired during the experiments in which the crowns were pulled underneath the water surface. **c, d** Top- and side-view video images of experiments involving the pulling of artificial substrates perpendicularly from underneath the water surface. Note that the central regions of these substrates (single asterisk) contained water. Overlaid white dotted lines indicate the free water surface in **a–d**. Red arrowheads indicate the positions of the crowns in **a** and **b** and the artificial substrates in **c** and **d**, respectively.

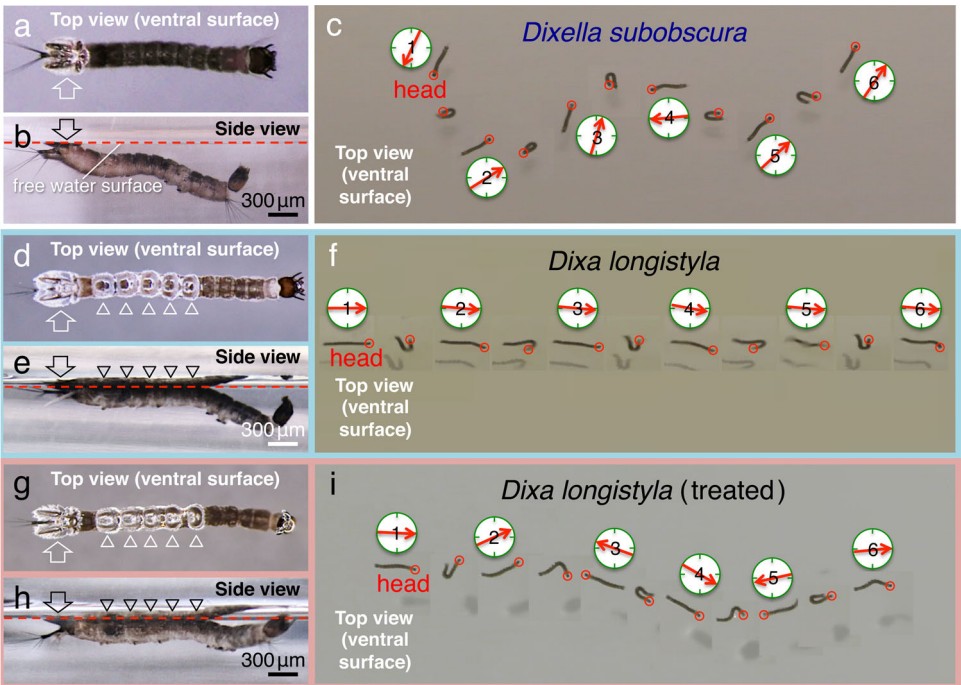

**Fig 6 Observations of the larvae of *Dixella subobscura* and the larvae of *Dixa longistyla* in swimming behaviour.** Comparison of the control larva of *Dixella subobscura* in **a–c**, the control larva of *Dixa longistyla* in **d–f** and the *Dixa* larva in which the crowns were treated with sunflower seed oil in **g–i**. **a, d, g** Top views. **b, e, h** Side views. **c, f, i** Images of the direction during the swimming movement of the larvae. The directions of the red arrows indicate the forward orientation of the larvae.

formulations[18]. When the hydrophilic regions of all five crowns were treated with the oil, which changed the water contact angles ($\theta$) to 120° (Fig. 7a–c versus g–i), the forward–oriented swimming movement (Fig. 7d) was completely inhibited (Figs. 6g–i and 7j, and Supplementary Movie 10). Even after the treatment, their ability to float was maintained as if in natural conditions (Fig. 6g, h); however, further observations revealed that the hind segments slipped and/or rotated, resulting in a random swimming direction (Fig. 7k). The rotation of the hind segments was significantly greater in the treated larvae (Fig. 7l) (86.9 ± 19°, mean ± S.D., $N = 20$) than in the control larvae (Fig. 7e, f) (16.5 ± 11°, mean ± S.D., $N = 20$) ($P < 0.01$). In the treated larvae, although the morphological features of the hydrophilic structures were unchanged, oil was observed on and around the fine structures via electron microscopy (Supplementary

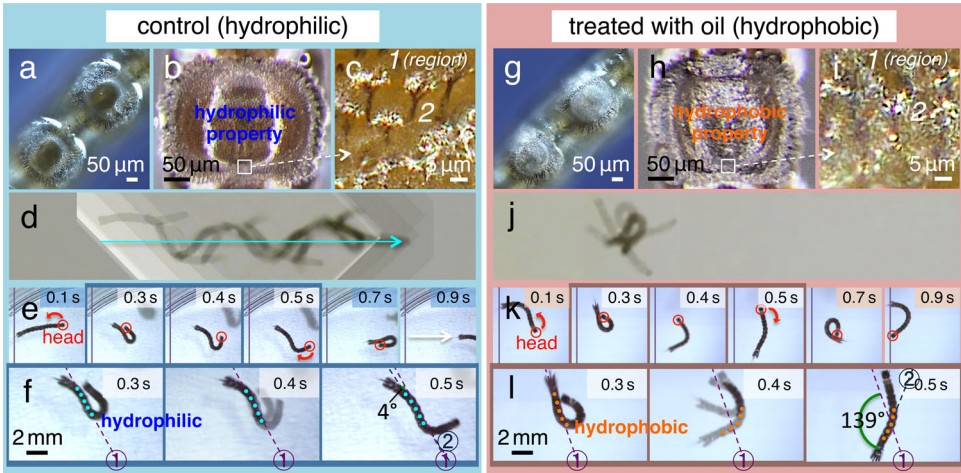

**Fig 7 Analysis of the crown structures in swimming behaviour. a–c**, **g–i** Crowns in the hind segment of the ventral surface, in the control larva in **a–c** and in the specimens treated with sunflower seed oil in **g–i**. **d**, **j** Merged sequential images taken during the swimming movements of the control larva in **d** and the larva treated with the oil in **j**. **e**, **f**, **k**, **l** Sequences of high-magnification images from time-lapse videos of movements of the control larva in **e** and **f** and the larva treated with the oil in **k** and **l**.

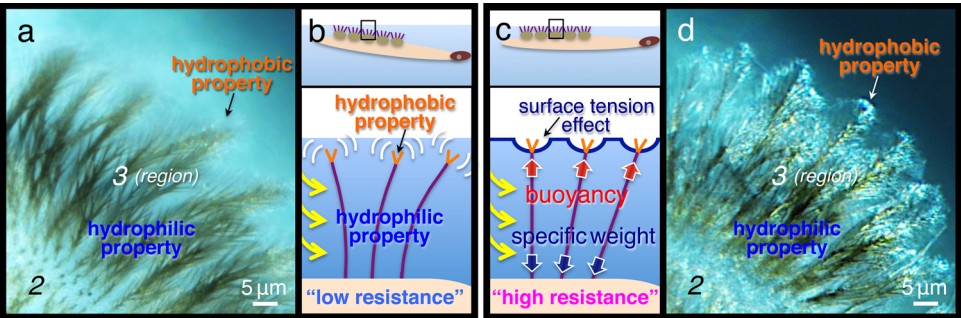

**Fig 8 Microscopy images and schematics depicting the combination of hydrophobic and hydrophilic properties in the crown causing the high resistance condition during larval locomotion. a**, **d** High-magnification differential interference microscopy images of the tips of the hairy crown structures and **b**, **c** schematic drawings. Image of the flexible hairy structures away from the water surface in **a** and **b**. Note that, with this condition, the hydrophobic hairy structures are unable to resist the water flow (yellow allows). When the hydrophobic hairy tips are attached to the water surface, both ends of the hairy structures are pulled by the surface tension and the specific weight of the larva in **c** and **d**. These fixed hydrophilic hairy structures produce high resistance against water flow during locomotion.

Fig. 2a–c versus d, e), and an additional air layer was visible in light microscopy images (Fig. 7g–i). In contrast, the oil and the air layer were obviously lost after 1 h (Supplementary Fig. 2f), and the ability to swim straight in the forward direction was recovered. These results suggest that the hydrophilic property of the five-crown structures in the posterior segments supports the larva in orienting its movement beneath the water surface (cf. Fig. 8 in Discussion).

## Discussion
We reported the remarkable ability of Dixidae larvae to use hydrophobic hairy structures on the ventral side of the abdomen to attach to a water surface and reside just below it (Figs. 1–5). The morphological effect of hydrophobic structures has been previously examined in various water beetles[1, 19,20]. The effect can be explained by two proposed hypotheses: (1) The roughness of the structures increases the surface area, which leads to the hydrophobic property (Wenzel model)[10]. (2) Below the water, air can remain within the rough structures, enhancing hydrophobicity (Cassie–Baxter model)[11]. As per these principles, in some aquatic insects, the assembly of comparatively small air bubbles, trapped on the surfaces of fine hairy structures, provides the buoyant force required for floating[21,22].

In the present study, we found that Dixidae larvae did not rely on such air bubbles to float. This was because, when the water surface tension was reduced, the hydrophobic hairy structures in the crown promptly lost their floating ability (Fig. 4b). This result was similar to that of an experiment with water striders. Striders have superhydrophobic hairy structures on their legs, which they use to float on water[3,4]. They are unable to float on water that has reduced surface tension. In the present study, we discovered that, once the tips of the hairy surfaces were attached to the water surface, air was only observed at these contact points (Fig. 2a–c, e and Supplementary Movies 1, 3 and 4), showing firm adhesiveness to the water surface (Fig. 5a, b; and Supplementary Movies 6 and 7). Presumably, this was because, with a large number of hydrophobic hairy tips repelling water at the water surface, the surface tension effect was generated by numerous point contacts (Fig. 2a, b, cf. Fig. 8). The assembly of comparatively tiny hairy tips causing firm adhesiveness is very similar to the setae of a gecko[23] or an insect's foot[24], which adheres to smooth vertical surfaces. In geckos, however, the adhesion is believed to be due to van der Waals forces.

In addition to the surface tension effect, we assume that the direct interaction between the hydrophobic hairy tips and air may also enhance adhesiveness. Pushkarova and Horn investigated the

adsorption of air bubbles on substrates and found that air bubbles adhere strongly to hydrophobic surfaces and weakly to hydrophilic surfaces[25]. Our experiments using artificial substrates also showed this tendency. When artificial substrates attached to the water surface were pulled perpendicularly into the water, the substrates with a hydrophobic surface allowed air to follow deeper into the water (Fig. 5c, d). On the basis of these results, we suggest that the complex of hydrophobic hairy structures in the crown acts as an adhesive disc to the water surface, binding the animal to the water surface using the surface tension and/or air as a binding agent. Thus, Dixidae larvae are able to cling to the water–air interface.

We also found that the crowns have a different function. Our results revealed that, by using the resistance of water, the hydrophilic region in the crown, including the super-hydrophilic centre structures, acts as a rudder during locomotion (Figs. 6d–f and 7a–f). To achieve this function, once the tips of the flexible crown hairs (Fig. 3) are attached to the water surface (Fig. 8a, b), the tensile strength is generated entirely in the fixed hairs between the surface tension and the specific weight of the larva (Fig. 8c, d). The phenomenon increases the water resistance in the hydrophilic region, which becomes a firm barrier at a large surface area in the crowns. However, when the hydrophilic regions are converted into hydrophobic ones, an additional air layer appears (Fig. 7g–i), and this might counteract the effect of the rudder[19]. As a consequence, when the larvae move, the swimming trajectory is no longer a straight line (Figs. 6g–i and 7g–l).

As we have shown above, without the reciprocal regulation of the hydrophobic and hydrophilic properties in the crowns, the creatures would not be able to remain beneath the water surface (Fig. 4b) or swim naturally (Figs. 6g–i and 7g–l). Based on the present investigation, it is suggested that the combination of opposing hydrophobic and hydrophilic properties is essential for the stabilisation of these larvae beneath the water surface. It has recently become very difficult to obtain a sufficient number of these creatures for our experiments, presumably because of global warming and environmental pollution. We believe that it is necessary to conserve such unique species for bioinspiration because their contrasting adjacent biological surface features are extremely relevant, not only to biological fields but also to various other scientific disciplines.

## Methods

**Animals**. By sweeping the surface of the water with a tropical fishnet, larvae of *Dixa longistyla* (Takahashi, 1958, Diptera: Dixidae) were collected from a pond at the National Museum of Nature and Science, Tsukuba (140°06′E, 36°06′N) and a stream near Lake Sanaru, Hamamatsu (137°41′E, 34°42′N). Larvae of *Dixella subobscura* (Takahashi, 1958, Diptera: Dixidae) were also collected from a stream near Lake Sanaru, Hamamatsu (137°41′E, 34°42′N). Prior to use in the experiments, the collected organisms were spread onto a shallow dish and manipulated with a small pipette by using a head-mounted binocular loupe.

**Measurement of specific weight**. The specific weight of the larvae was measured using an electronic hydrometer (SD-200L, Alfa Mirage Co. Ltd.), following the manufacturer's instructions.

**Reducing the surface tension of the liquid**. The surface tension of the water was modified by the addition of an amphiphilic compound, polysorbitan monolaurate (Tween 20; Wako Pure Chemical Industries). In the experiment, 10 mL of 1% (v/v) Tween 20 solution, dissolved in distilled water, was gently poured into 90 mL of distilled water (0.1% Tween 20 as a final concentration). The larvae floated on the surface of this solution (Fig. 4b).

**Recording movements**. Top and side views of the movements of the specimens were recorded with a digital camera (Olympus E-5, in video capture mode) and a 1000-fps high-speed camera (HAS-220C, DITECT), with a variable lighting system comprising an LED illumination device (KOIZUMI TR.).

**Electron microscopy**. Field-emission scanning electron microscopy (FE-SEM) was carried out with a JEM-7100F (JEOL) and/or a Hitachi S-4800 instrument, operated at an acceleration voltage of 1.0 kV. The observation chamber was maintained at a vacuum of $10^{-3}$–$10^{-6}$ Pa. Secondary electrons were accumulated by a lower detector within the instrument in each case. Other experimental parameters are as follows: working distance = 8 mm; aperture size = 100 μm; scan speed = 10–15 fps (for both SEM instruments).

**FE-SEM sample preparation for observing living specimens**. The NanoSuit method[13–15] was used to observe the intact surface structure of the larvae (Supplementary Movie 2). Specimens were placed directly into the SEM instrument without any pre-treatment such as chemical fixation, dehydration, or an ultrathin, electrically conducting material coating. A NanoSuit was formed following electron beam irradiation in the SEM instrument.

**Preparation of samples using conventional fixation for SEM**. For conventional SEM observations, samples were prefixed with 2% glutaraldehyde in 0.1 M phosphate buffer (pH 7.4) and postfixed in 1% $OsO_4$ in the same buffer. The samples were then dehydrated using a graded series of ethanol, transferred to t-butyl alcohol, freeze-dried (JFD300, JEOL) and coated with an ultrathin layer of $OsO_4$ (PMC-5000, Meiwa).

**Quantitative surface hydrophobicity measurement using a contact angle metre**. The interfacial tensions arising from the interaction of different surfaces with water were compared in terms of the surface wettability measured using automatic microscopic contact angle metres (for the artificial substrates, Kyowa DMs-401, and for the larva specimens, Kyowa MCA-3). A 1-μL or 1-nL droplet of water was formed on the tip of a glass capillary with an inner diameter of 0.5 mm (Kyowa DMs-401) or 5 μm (Kyowa MCA-3), respectively. A side view of the droplet interacting with the surface was captured with a high-speed camera (1000 fps; HAS-220C, DITECT). Water contact angles ($\theta$) were measured from the captured images[17].

**Pull-off experiments and preparation of artificial substrates**. To examine the effects of substrate–liquid interface tension, we conducted experiments to pull specimens attached to the water surface down into the water. The bodies of living larvae were fixed onto metal clips by double-sided tape and then pulled perpendicularly down into the water at a constant speed (30 μm/s) using a Water Robot Micromanipulator (Narishige Japan). The depth at which the specimen detached from the water surface was determined (Fig. 5a, b).

We also conducted this experiment using artificial substrates (Fig. 5c, d), which had surfaces with different wettabilities, i.e., different levels of hydrophobicity and hydrophilicity. We coated nickel-ring plates (6.0-mm outer diameter, 2.0-mm inner diameter and 0.4-mm thickness) with gold using a standard ion-sputtering device (JFC-1100, JEOL) for 10 min at a vacuum of 1.0 Pa, a DC voltage of 1.0 kV (8.0 mA), and a temperature of 20 °C. To produce two types of surface-modified plates with different interfacial free energies, the gold-coated plates were immersed in an ethanol solution containing 1-mM 11-mercapto-1-undecanol (hydrophilic) or 1-dodecane thiol (hydrophobic) for 12 h. After the surface treatments, the chemically modified plates were rinsed with ethanol and dried overnight[17]. These ring plates were pulled below the water surface at a constant speed (200 μm/s), and the depths at which separation from the water surface occurred were determined.

**Crown treatment with oil and body-orientation measurement during swimming**. The wettability of the hydrophilic region in the five crowns, on the ventral side of the hind segments, was modified using sunflower seed oil such that the region became hydrophobic ($\theta = 120°$) (Figs. 6g–i and 7g–l, and Supplementary Fig. 2d–f). The experimental larvae were picked out from the water and blotted briefly on dry filter paper. A small amount of oil was gently applied to the hydrophilic region around the central structures using the tips of a pair of forceps. The treated specimens were washed with distilled water twice to remove excess oil and then placed back in water for the experiments.

During the swimming motion of the larvae, the orientation of the hind segments relative to the direction of motion was determined. To determine the orientation of the hind segments, a line was drawn through the centres of the five crowns (Fig. 7f, l). Results from control larvae and specimens treated with the oil were compared.

**Statistics and reproducibility**. Statistical analysis was carried out using Student's $t$ test. The tests were two-tailed, and the actual $P$ value for each test was generated with the significance level set at $P < 0.01$.

**Reporting summary**. Further information on research design is available in the Nature Research Reporting Summary linked to this article.

## Data availability

The authors declare that all data generated or analysed during the study are stored and securely backed-up. In addition, the original datasets including the image sets in the main figures and any remaining information are available from the corresponding authors on reasonable request.

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

## Acknowledgements

We wish to express our thanks to two anonymous referees for helpful suggestions and comments on the manuscript, and to Dr. T. Nakamura for identifying experimental animals. We are grateful to Dr. H. Kawasaki for helpful suggestions. This work was supported by Grants-in-Aid for Scientific Research for Y.T. (JP25292198, 17K08150, 20K06071); a Grant-in-Aid for Challenging Exploratory Research for Y.T. (15K14558); and Grants-in-Aid for Scientific Research in Innovative Areas ('Innovative Materials Engineering Based on Biological Diversity') for T.H., M.S. (JP24120001, JP24120004) and Y.T. (15H01598).

## Author contributions

C.S., Y.T., H.S. and T.H. planned the experiments. C.S., Y.T., H.S., D.I., T.S., S.N., M.S. and T.H. performed the experiments and analysed the data. C.S., Y.T. and T.H. prepared the manuscript.

## Competing interests

The authors declare no competing interests.
