## [Peer Review File · Communications Biology]

Reviewers' comments:

Reviewer #1 (Remarks to the Author):

The authors present the results of experiments on the larvae of *Dixidae*, or meniscus midges. They ask the question of how the insect attach to the water surface. The authors give a qualitative answer: the five adhesive pads on the larvae have hydrophobic and hydrophilic parts that prevent detachment from the water surface. They conduct some creative experiments to "knock out" the different regions, and also build a mimic to demonstrate the general principle. Overall, I believe their general result is correct, but there are some details missing. Tthe authors need to improve the writing in places to clearly explain their findings. While this is mainly a descriptive study with no theoretical analysis, I can see this paper inspiring future work.

I like how the authors conduct "knock-out" experiments by adding oil to prevent the hydrophilic parts from working, which show that the hydrophilic parts are necessary for swimming in a straight line. The mechanism to me is not clear however. They state that the hydrophilic parts act like a rudder, but their meaning is not clear. A schematic should be given. Also, what happens when only a fraction of the five attachments are knocked out?

Why are there five attachment points, and what is the significance of their position on the body? Is swimming made possible with the tail or with undulations of the whole body. Overall the swimming videos were not at a high enough frame rate. A high speed camera should be used.

The figures are useful, but there are too many movies with this paper. I think the detachment videos 6867_0_video_171194_qjml56 are the most interesting.

The experiment showing that the hair tips are hydrophobic and the rest of the hairs are hydrophilic is interesting. The experiments confirming this with different regions of hair are nice. How does a single hair change from hydrophobic to hydrophilic with distance along shaft?

The writing could be improved. In a number of places the authors use the words "surface tension," but they should really say capillary force.

I like the use of the disc mimics, but I feel the authors did not go very far to mimic the insect. For instance, its not clear why the insects needs both hydrophobic and hydrophilic regions. The authors seem to say that a hydrophobic region is enough.

The section about swimming I found to be a little incomplete. Its not clear to me why attachment improves swimming. Does it help with stability? It seems like attachment would incur extra resistance to forward motion. To do this experiment, maybe the authors need to compare swimming with attachment to water surface to swimming in a thin layer of water. In both cases, the insect may maintain stability.

Other insects also attach to the water surface, such as mosquito larvae. Those citations should be included. I think such larvae can also propel themselves while attached.

Reviewer #2 (Remarks to the Author):

The manuscript "Hydrophobic/hydrophilic "crown" structure allows aquatic insects to reside effectively beneath the water`s surface" by Suzuki et al. describes the functional morphology of the crowns located on the ventral side of the abdomen of the larvae of *Dixidae*, an aquatic Dipteran family whose larvae typically live in lentic freshwaters just beneath the water surface film. The investigation was performed through observations under light and scanning electron microscopy, contact angle measurements, pull-off experiments. Experiments with artificial substrates of different wettability were also performed. The research is well performed and the results are interesting and not only shed light on a peculiar adaptation of *Dixidae* living in water

attached to the surface film at the interface air-water, but they can have also interesting implications for biomimetic purposes.

I recommend publication on *Communication Biology* after revision.

Introduction

-lines 43-48, not clear, why do you relate insect diving ability only to feeding and predator avoidance?

Lines 53-57 When you present the insect object of your investigation you can add some more general information about Dixidae. Maybe you can introduce here the crown which I guess are important diagnostic characters for these insects. Moreover, you can move here the first lines of the Results and Discussion (lines 62-67) with citation 9.

Somewhere in the introduction you could cite the following manuscript reporting in mosquito larvae a hydrophobic wettability of the syphon in order to float:

Lee SC, Kim JH, Lee SJ. Floating of the lobes of mosquito (*Aedes togoi*) larva for respiration. *Sci Rep.* 2017;7:43050. Published 2017. doi:10.1038/srep43050

Results and Discussion

-line 68 Observations with higher magnification. With light microscopy? Please, specify.

-lines 73-74 Please, move to material and methods and simply report the results and discussion in this section. This problem is present in different points in the results and discussion section.

Please, move all the details regarding mat and met in the proper section.

-line 79 Please, show wider pictures (especially SEM details of fig. 2e-l) and try to describe in more detail the morphology of the structures.

-line 81 sparse hairs: are these structures hairs? I have the impression that they are not hairs but branched structures. Show bigger images and describe them more precisely.

-line 109-110 Please, move to mat and met section and report here only the results

If possible, the pulling experiments would be more detailed and elegant measuring the force with a force transducer

We are grateful to the reviewers for their critical comments and useful suggestions, which have helped us improve our manuscript. We have taken all the comments and suggestions into account in the revision of our manuscript.

Moreover, to explain our findings clearly, we have changed the manuscript type to 'full paper' and this included 'Results' and 'Discussion' sections separately. The reviewer comments are followed by our responses (italicized) and a list of changes made to the main text.

*We hope that the revised manuscript is suitable for publication in **Communications Biology**.*

Reviewers' comments:

Reviewer #1 (Remarks to the Author):

The authors present the results of experiments on the larvae of dixidae, or meniscus midges. They ask the question of how the insect attach to the water surface. The authors give a qualitative answer: the five adhesive pads on the larvae have hydrophobic and hydrophilic parts that prevent detachment from the water surface. They conduct some creative experiments to "knock out" the different regions, and also build a mimic to demonstrate the general principle. Overall, I believe their general result is correct, but there are some details missing. Tthe authors need to improve the writing in places to clearly explain their findings. While this is mainly a descriptive study with no theoretical analysis, I can see this paper inspiring future work.

I like how the authors conduct "knock-out" experiments by adding oil to prevent the hydrophilic parts from working, which show that the hydrophilic parts are necessary for swimming in a straight line. The mechanism to me is not clear however. They state that the hydrophilic parts act like a rudder, but their meaning is not clear. A schematic should be given. Also, what happens when only a fraction of the five attachments are knocked out?

We thank the Reviewer #1 for their positive comments on the 'knock-out' experiments. To explain the mechanism of swimming in a straight line and why the hydrophilic parts act like a rudder, we have added a schematic to Figure 8 in the revised version.

We thank the reviewer for the helpful idea of knocking out only a fraction of the five attachments. We assume that including such data would allow readers to understand the role of hydrophilic parts of the attachments. Unfortunately, we are currently unable to obtain living samples to conduct further experiments because Dixa larvae are seasonal. We have already taken five seasons to conduct the experiments reported in the manuscript, and the season of Dixa larvae starts around June and ends in October. Furthermore, it has recently become very difficult to find a sufficient amount of larvae for experiments, presumably because of global warming and environmental pollution. Under these circumstances, to show how the hydrophilic parts in the crowns are necessary for swimming in a straight line, we have incorporated some extra data in the revised version of the manuscript (please see revised Figure 6). The addition shows the swimming movements of larvae of a different species (Dixella subobscura), which belongs to the same Dixidae family but does not have the crown structures on the ventral side of the hind segments. These Dixella larvae showed random swimming directions, while the Dixa larvae swam in an almost straight line. By further comparing these results to the "knock-out" experiments of the Dixa larvae we had already performed, we hope that the manuscript clarifies how the hydrophilic parts act like a rudder. Furthermore, to explain these results clearly, we have revised the text (Lines 144-182 on Pages 8-9).

Why are there five attachment points, and what is the significance of their position on the body? Is swimming made possible with the tail or with undulations of the whole body. Overall the swimming videos were not at a high enough frame rate. A high speed camera should be used.

Thank you for suggesting this possibility. We are planning to conduct further experiments to clarify this point in the next season, provided we are able to collect larvae. We will also focus on developing some biomimetic devices. We would like to report these precise mechanisms separately.

Following the suggestion, we have re-edited the swimming videos to achieve better frame rates. Even though the specimens were very small, low-magnification recording with a stereomicroscope was possible. However, it was very difficult to record these movements with high magnification by using an upright microscope because it required a very strong light source. Please refer to re-edited Supplementary Movies 9 and 10.

The figures are useful, but there are too many movies with this paper. I think the detachment videos 6867_0_video_171194_qjml56 are the most interesting.

As per the suggestion, we have reduced the number of the Supplementary Movies from thirteen to nine but included a new one to show the real movement, as shown in Figure 6 in the revised manuscript. However, if the number of videos should be reduced further, we will attempt to comply.

The experiment showing that the hair tips are hydrophobic and the rest of the hairs are hydrophilic is interesting. The experiments confirming this with different regions of hair are nice. How does a single hair change from hydrophobic to hydrophilic with distance along shaft?

In these experiments, individual hairy structures were too small to conduct measurements of wettability with respect to distance along the shaft. In the present

investigations, we were limited to the technique of using a microscopic contact angle meter to measure the wettability of an assembly of hairs; even this technique used a very small, 1-nL droplet of water.

The writing could be improved. In a number of places the authors use the words "surface tension," but they should really say capillary force.

We are grateful for this comment. As you pointed out, this phenomenon could be largely related to capillarity. Indeed, if the space between hairy structures is sufficiently small, the combination of surface tension and the adhesive force among the hairs acts to draw down the liquid. However, at present, although the larvae were prevented from adhering to the water surface in our experiment where the surface tension was reduced, we have not yet been able to measure the direct relationship among the size, hair-like structures, and pulling force. Thus, we would like to retain the usage of "surface tension" when describing this phenomenon.

I like the use of the disc mimics, but I feel the authors did not go very far to mimic the insect. For instance, its not clear why the insects needs both hydrophobic and hydrophilic regions. The authors seem to say that a hydrophobic region is enough.

Thank you for this important comment. In the revised manuscript, we have added investigations on the hydrophobic–hydrophilic properties of the crown structure, as shown in Figures 1-3 in the revised manuscript. The role of the hydrophobic region is confirmed in Figures 4 and 5 (the disc mimics in Fig. 5c, d show the function of the hydrophobic regions). The role of the hydrophilic region is examined in Figures 6 and 7 (the experiment using the artificial substrate in Supplementary Fig. 2 reveals the function of the hydrophilic regions). We have summarized the hydrophobic–hydrophilic interaction in Figure 8. Further, we have revised the text and legend of Figure 8 as follows:

(Lines 224-230 on Pages 11-12)

“To achieve this function, once the tips of the flexible crown hairs (Fig. 3) are attached to the water surface (Fig. 8a), the tensile strength is generated entirely in the fixed hairs between the surface tension and the specific weight of the larva (Fig. 8b). This increases the hydrophilic resistance, which becomes a firm obstacle (Fig. 8c). However, when the hydrophilic regions are converted into hydrophobic ones, there appears to be a weakened affinity to water (Fig. 8d). As a consequence, when the larvae move, the swimming trajectory is no longer a straight line (Figs. 6g-i, 7g-l).”

(Lines 357-365 on Page 22)

“Schematic drawings of the hydrophilic structures in the crown causing high resistance during larval locomotion. (a) Image of the flexible hairy structures away from the water surface. Note that, with this condition, the hydrophobic hairy structures are unable to resist the water flow (yellow allows). (b) When the hydrophobic hairy tips are attached to the water surface, both ends of the hairy structures are pulled by the surface tension and the specific weight of the larva. (c) These fixed hydrophilic hairy structures with a high affinity to water produce high resistance against water flow during locomotion. (d) On the other hand, when the hydrophilic regions are converted into hydrophobic regions, a weakened affinity to water results in low resistance.”

The section about swimming I found to be a little incomplete. Its not clear to me why attachment improves swimming. Does it help with stability? It seems like attachment would incur extra resistance to forward motion. To do this experiment, maybe the authors need to compare swimming with attachment to water surface to swimming in a thin layer of water. In both cases, the insect may maintain stability.

We apologize for the confusion. We have revised the manuscript to explain why attachment improves the swimming movement, as mentioned above. We consider that the hydrophilic resistance in the crown enables the posterior segments to be stable

under the water surface, which prevents insufficient rotation during the movement. Even though this would cause some extra resistance to forward motion, this specific property supports the larva in orienting its movement beneath the water surface. As suggested, an additional experiment in a thin layer of water might be helpful for examining the direct interactions of these resistances; we will consider performing such an experiment in future research.

Other insects also attach to the water surface, such as mosquito larvae. Those citations should be included. I think such larvae can also propel themselves while attached.

As mentioned above, in the revised manuscript, we have reported additional experiments using mosquito larvae of a different species; these experiments show how these larvae propel themselves under the water surface. The reference concerning to other mosquito larvae residing under water has also been cited in the introduction as follows:

(Lines 45-46 on Page 3)

“Furthermore, the floating for respiration has been examined in Aedes mosquito larvae in water⁷.”

Reviewer #2 (Remarks to the Author):

The manuscript “Hydrophobic’hydrophilic “crown” structure allows aquatic insects to reside effectively beneath the water`s surface” by Suzuki et al. describes the functional morphology of the crowns located on the ventral side of the abdomen of the larvae of Dixidae, an aquatic Dipteran family whose larvae typically live in lentic freshwaters just beneath the water surface film. The investigation was performed through observations under light and scanning electron microscopy, contact angle measurements, pull-off experiments. Experiments with artificial substrates of different wettability were also performed. The research is well performed and the results are interesting and not only shed light on a peculiar adaptation of Dixidae living in water attached to the surface film at the interface air-water, but they can have also interesting implications for biomimetic purposes.

I recommend publication on Communication Biology after revision.

Introduction

-lines 43-48, not clear, why do you relate insect diving ability only to feeding and predator avoidance?

Thank you for raising this critical question. As per your suggestion, we have revised the Introduction section (second paragraph of Page 3) as follows:

“By contrast, other aquatic organisms possess hydrophilic properties that enable them to live within a water column. For example, the feeding habits of various diving beetles in water have been investigated^{5,6}. Furthermore, the floating for respiration has been examined in Aedes mosquito larvae in water⁷. Anopheline larvae usually dive under water and remain at the bottom for some time. This pattern of behaviour suggests that their diving ability may be related to feeding and predator avoidance⁸.”

Lines 53-57 When you present the insect object of your investigation you can add some more general information about Dixidae. Maybe you can introduce here the crown which I guess are important diagnostic characters for these insects. Moreover, you can move here the first lines of the Results and Discussion (lines 62-67) with citation 9.

We appreciate this helpful suggestion. We have revised the Introduction section (third paragraph of Page 3) as follows by moving here the first few sentences of the Results and Discussion (in the previous manuscript) with the citation:

*“In contrast to these organisms, the larvae of Dixidae, a family of aquatic Nematocera flies (Diptera), exhibit a unique characteristic^{9,10}. The larvae of Dixidae usually live along the margin of floating branches, stalks of waterweed, rocks in a pond, or parts of a stream where the current is very gentle. Like mirror images of water striders, the larvae of Dixidae lie with their ventral surface just under the water surface. They never dive into the water and are never completely submerged within the water column; therefore, they are known as meniscus midges. Little is known, however, about how the larvae are able to live beneath the water surface. In this study, we used larvae of *Dixa longistyla* (Diptera: Dixidae) as experimental specimens. We examined their ability to live just below the water surface, focusing on crown-like structures on the ventral side of the hind segments¹⁰. We provide evidence that a combination of hydrophobic and hydrophilic structures enables the larvae to reside in such an ecological niche.”*

Somewhere in the introduction you could cite the following manuscript reporting in mosquito larvae a hydrophobic wettability of the syphon in order to float:

Lee SC, Kim JH, Lee SJ. Floating of the lobes of mosquito (*Aedes togoi*) larva for respiration. *Sci Rep.* 2017;7:43050. Published 2017. doi:10.1038/srep43050

As described in a previous response, the reference has been cited in the introduction as follows:

(Lines 45-46 on Page 3)

“Furthermore, the floating for respiration has been examined in Aedes mosquito larvae in water⁷.”

Results and Discussion

-line 68 Observations with higher magnification. With light microscopy? Please, specify. Please accept our apologies for the confusion. We have added an explanation as follows:

(Lines 68-69 on Page 4)

“High-magnification observations with light microscopy show hairy structures in the crowns.”

-lines 73-74 Please, move to material and methods and simply report the results and discussion in this section. This problem is present in different points in the results and discussion section. Please, move all the details regarding mat and met in the proper section.

According to the suggestion, we have revised the sentence as follows:

(Lines 73-74 on Page 4)

“Observations with electron microscopy (Supplementary Movie 2)¹³⁻¹⁶ show that the crowns consist of three different structures (Fig. 2g-l, Supplementary Fig. 1).”

-line 79 Please, show wider pictures (especially SEM details of fig. 2e-l) and try to describe in more detail the morphology of the structures.

We have revised Figure 2 with a better SEM image. Moreover, we have included additional SEM images in the revised version of Supplementary Figure 1. We have also modified sentences as follows:

(Lines 74-80 on Pages 4-5)

“In the centre (region 1), flat leaf-like structures ~3-5 μm in width are observed with central folds (Fig. 2i). Surrounding the central structures (region 2), dense hairs

(microtrichia) less than 0.3–1 μm in diameter are detected (Fig. 2j). On the outer periphery (region 3), sparse hairs (protrusion) \sim 0.5–2 μm in diameter are observed (Fig. 2k). In regions 2 and 3, the tips of the hairs are further divided into branched structures and can be distinguished by their diameter of 0.1 μm (Fig. 2d, j) or 0.3 μm (Fig. 2f, k), respectively.”

-line 81 sparse hairs: are these structures hairs? I have the impression that they are not hairs but branched structures. Show bigger images and describe them more precisely.

Thank you for pointing out this issue. According to the suggestion, we have revised Figure 2f, k and Supplementary Figure 1c, i with larger images, as already mentioned above. We recognized that these structures were thin and long, showing flexibility as well (please see revised Figure 3). Based on their specific features, we had described them as hair-like structures. We have now checked how similar structures were described in the previous reports. In References 3 (Hu et al, 2003) and 4 (Gao and Jiang, 2004), for example, the authors call the fine structures on the surface of the leg of water striders ‘hairs’, although they are not as thin or long as the fine structures on Dixa larvae. Therefore, we would like to retain the usage of ‘hair’ in the present paper. Furthermore, following your suggestion, we have added the following description:

(Lines 78-80 on Page 5)

“In regions 2 and 3, the tips of the hairs are further divided into branched structures and can be distinguished by their diameter of 0.1 μm (Fig. 2d, j) or 0.3 μm (Fig. 2f, k), respectively.”

-line 109-110 Please, move to mat and met section and report here only the results

We have moved these sentences to the Methods section and revised the sentences as follows:

(Lines 107-110 on Page 6)

“To address this issue, we designed a simple experiment. When the liquid surface

tension was reduced, the crown structures lost contact with the water surface, and the position of the larvae quickly changed from being parallel to being perpendicular to the surface (Fig. 4b)."

If possible, the pulling experiments would be more detailed and elegant measuring the force with a force transducer

Thank you for raising the valuable point of measuring the force with a force transducer. At present, it appears that the method requires considerable preparation; therefore, we would like to attempt the relevant modifications separately in a new report.

Reviewers' comments:

Reviewer #1 (Remarks to the Author):

I like the new experiments conducted with the *Dixella* larvae. They make clear the importance of the attachment zones. The writing is also improved. My main concern is with the proposed mechanism of drag due to the hydrophobic regions. I also have two minor comments.

Fig 2-3. The authors use a blue, red and purple color scheme for the hydrophobic, hydrophilic and super hydrophobic parts. The issue is that the red and purple are just too similar. Also red triangles are used elsewhere. Can they use other colors? Or maybe dotted lines?

Fig 6c should be labelled *Dixella subscura*. The *Dixa* name confused me a little.

Fig 8c-d are confusing and likely to be incorrect. I suggest deleting. When hydrophilic structure is completely immersed in water, its drag resistance is no different than a hydrophobic structure. It is only when there is an air layer that the drag resistance might be different.

I believe that the attachment to the water surface simply reduces the degrees of the freedom and keeps their motion more planar. I suggest the authors keep into account other possible mechanisms for how the swimming is improved with surface attachment. I suggested last time that a thin layer of water would do the same. In any case, the speculation about "hydrophilic resistance" is not supported.

162- -- the words "hydrophilic resistance" are not used in literature and I don't understand what the authors mean. Those parts and Fig 8c-d should just be deleted. Its pure speculation and doesn't really make sense.

It is unfortunate that the larvae are not able to be found because of changing global conditions. Perhaps that point can be made in discussion and used to make a point to conserve species for bioinspiration.

We are grateful to Reviewer #1 for the additional comments and useful suggestions, which have helped us to further improve our manuscript. We have taken all the comments and suggestions into account in the revision of our manuscript. The reviewer's comments are followed by our responses (italicized) and a list of changes made to the main text. We hope that the revised manuscript is suitable for publication in Communications Biology.

Reviewers' comments:

Reviewer #1 (Remarks to the Author):

I like the new experiments conducted with the *Dixella* larvae. They make clear the importance of the attachment zones. The writing is also improved. My main concern is with the proposed mechanism of drag due to the hydrophobic regions. I also have two minor comments.

We thank Reviewer #1 for their positive comments on the revision of our manuscript. Following their suggestions, we have revised our manuscript.

Fig 2-3. The authors use a blue, red and purple color scheme for the hydrophobic, hydrophilic and super hydrophobic parts. The issue is that the red and purple are just too similar. Also red triangles are used elsewhere. Can they use other colors? Or maybe dotted lines?

We have changed these colors in Figures 2 and 3. In the revised manuscript, we use orange instead of red (please see revised Figures 2 and 3).

Fig 6c should be labelled *Dixella subscura*. The *Dixa* name confused me a little.

*We apologize for the confusion. We have corrected the name to *Dixella subscura* in revised Figure 6c.*

Fig 8c-d are confusing and likely to be incorrect. I suggest deleting. When

hydrophilic structure is completely immersed in water, its drag resistance is no different than a hydrophobic structure. It is only when there is an air layer that the drag resistance might be different.

Thank you for making this important point. Following your suggestion, we have deleted Figure 8c, d and the related discussion, and further improved Figure 8. In the revised manuscript, we have also added some explanation about the air layer formed on the treated specimens, as shown in Figure 7 (compare revised Figure 7a and g), and improved the text as follows:

(Lines 177-182 on Page 9)

“In the treated larvae, although the morphological features of the hydrophilic structures were unchanged, oil was observed on and around the fine structures via electron microscopy (Supplementary Fig. 2a-c versus d, e), and an additional air layer was visible in light microscopy images (Fig. 7g-i). In contrast, the oil and the air layer were obviously lost after 1 h (Supplementary Fig. 2f), and the ability to swim straight in the forward direction was recovered”

I believe that the attachment to the water surface simply reduces the degrees of the freedom and keeps their motion more planar. I suggest the authors keep into account other possible mechanisms for how the swimming is improved with surface attachment. I suggested last time that a thin layer of water would do the same. In any case, the speculation about "hydrophilic resistance" is not supported.

As mentioned above, in this revised manuscript, we have added some explanation about the additional air layer (revised Figure 7g). Moreover, we have improved the text to include a description of a possible mechanism for the regulation of the swimming in the Discussion as follows:

(Lines 228-232 on Pages 11-12)

“The phenomenon increases the water resistance in the hydrophilic region, which becomes a firm barrier at a large surface area in the crowns. However, when the hydrophilic regions are converted into hydrophobic ones, an additional air layer appears (Fig. 7g-i), and this might counteract the effect of the rudder¹⁹.”

162- -- the words "hydrophilic resistance" are not used in literature and I don't understand what the authors mean. Those parts and Fig 8c-d should just be deleted. Its pure speculation and doesn't really make sense.

Please accept our apologies for the confusion caused by the use of this form of words. We have deleted the words "hydrophilic resistance" from the text, and from Figure 8c, d, as outlined in a previous response. We have also deleted Supplementary Figure 2 used in the previous version of the manuscript.

It is unfortunate that the larvae are not able to be found because of changing global conditions. Perhaps that point can be made in discussion and used to make a point to conserve species for bioinspiration.

We are very grateful for this suggestion. In the revised manuscript, we have added sentences to the Discussion as follows:

(Lines 239-244 on Page 12)

"It has recently become very difficult to obtain a sufficient number of these creatures for our experiments, presumably because of global warming and environmental pollution. We believe that it is necessary to conserve such unique species for bioinspiration because their contrasting adjacent biological surface features are extremely relevant, not only to biological fields but also to various other scientific disciplines."